# RECITAL: a non-inferiority randomised control trial evaluating a virtual fracture clinic compared with in-person care for people with simple fractures (study protocol)

Min Jiat Teng [1,2] Joshua R Zadro [1] Kristen Pickles [3] Tessa Copp [3] Miranda J Shaw,[2] Isabella Khoudair,[2] Mark Horsley,[4] Benjamin Warnock,[2] Owen R Hutchings,[2] Jeffrey F Petchell,[4] Ilana N Ackerman [5] Alison Drayton,[6] Rong Liu,[2,7] Christopher G Maher [1] Adrian C Traeger [1]

For numbered affiliations see end of article.

**Correspondence to**
Min Jiat Teng;
Min.Teng@health.nsw.gov.au

## ABSTRACT

**Introduction** Most simple undisplaced fractures can be managed without surgery by immobilising the limb with a splint, prescribing medication for pain, and providing advice and early rehabilitation. Recent systematic reviews based on retrospective observational studies have reported that virtual fracture clinics can deliver follow-up care that is safe and cost-effective. However, no randomised controlled trial has investigated if a virtual fracture clinic can provide non-inferior physical function outcomes compared with an in-person clinic for patients with simple fractures.

**Methods and analysis** 312 participants will be recruited from 2 metropolitan hospitals located in Sydney, Australia. Adult patients will be eligible if they have an acute simple fracture that can be managed with a removable splint and is deemed appropriate for follow-up at either the virtual or in-person fracture clinic by an orthopaedic doctor. Patients will not be eligible if they have a complex fracture that requires a cast or surgery. Eligible participants will be randomised to receive their follow-up care either at the virtual or the in-person fracture clinic. Participants at the virtual fracture clinic will be reviewed within 5 days of receiving a referral through video calls with a physiotherapist. Participants at the in-person fracture clinic will be reviewed by an orthopaedic doctor within 7–10 days of receiving a referral. The primary outcome will be the patient's function measured using the Patient-Specific Functional Scale at 12 weeks. Secondary outcomes will include health-related quality of life, patient-reported experiences, pain, health cost, healthcare utilisation, medication use, adverse events, emergency department representations and surgery.

**Ethics and dissemination** The study has been approved by the Sydney Local Health District Ethics Review Committee (RPAH Zone) (X23-0200 and 2023/ETH01038). The trial results will be submitted for publication in a reputable international journal and will be presented at professional conferences.

**Trial registration number** ACTRN12623000934640.

## STRENGTHS AND LIMITATIONS OF THIS STUDY

⇒ Pragmatic clinical trial embedded within two existing fracture clinics at two urban hospitals.
⇒ Measures hospital-level outcomes as well as patient outcomes and experiences.
⇒ Blinding of therapist or participants is not possible, although participants are blinded to the study hypothesis.
⇒ Methods and results from this trial may inform the evaluation of other virtual musculoskeletal services.
⇒ Study may not be sufficiently powered to determine subgroup effects, for example, based on specific fracture diagnosis.

## INTRODUCTION

In 2019, there were 178 million new fractures reported globally, an increase of 33.4% since 1990.[1] In Australia, the treatment costs of osteoporosis-related fractures were estimated to be $A2.34 billion in 2017.[2] With increasing numbers of people requiring care for their fractures, the burden on outpatient fracture clinics has also increased, causing long clinic wait times and productivity losses.[3 4] The recent pandemic further strengthens the requirement for health system efficiency.

Most simple fractures, including minimally displaced fractures, can be managed conservatively without surgery. These stable fractures are managed with short-term immobilisation, advice, pain relieving medication and early rehabilitation.[5] Traditional physical assessments at an outpatient clinic may not be required for conditions that have a clear prognosis and have been shown to recover well with conservative management.[6]

BMJ

Published studies have shown that virtual fracture clinics (VFCs) can manage patients with simple fractures.[7] Patients receive advice and management through phone calls and written handouts, rather than attending the outpatient clinic in person. Retrospective observational studies have reported that VFCs are associated with good patient satisfaction, increased cost efficiency for the hospital system, fewer adverse events and reduced presentations to in-person clinics.[8]

Despite a rise in VFCs since the recent pandemic, robust evaluations of their safety, effectiveness and cost-effectiveness are lacking. A recent systematic review of 21 publications suggested that VFCs could provide safe and cost-effective care to patients with acute fractures, though none of the included studies were randomised controlled trials.[8] It is currently unknown whether VFCs produce non-inferior outcomes compared with in-person care for patients with simple fractures.

We have designed a clinical trial to evaluate the effectiveness of a VFC for patients with simple fractures. The primary outcome of this trial is physical function at 12-week follow-up, measured using the Patient-Specific Functional Scale (PSFS). Secondary outcomes will include health-related quality of life, patient-reported experiences, pain, health cost, healthcare utilisation, medication use, adverse events, emergency department (ED) representations and surgery. A qualitative substudy will be conducted to explore the experiences, feelings and expectations of patients who use the VFC.

## METHODS AND ANALYSIS
### Design
The Fracture Clinic Trial (RECITAL) is a prospective two-arm, parallel group randomised controlled trial, using a non-inferiority design with nested economic and process evaluations. We chose a non-inferiority randomised controlled trial design as both study groups are existing hospital services, and the VFC is expected to have outcomes that are at least no worse than the in-person fracture clinic. This trial has been prospectively registered with the Australian New Zealand Clinical Trials Registry (ACTRN12623000934640). This document describes the trial protocol according to the Standard Protocol Items: Recommendations for Interventional Trials 2013 Statement.[9] Recruitment began in November 2023, with the final data collection expected to occur in November 2025.

### Setting
RECITAL will compare two existing models of care provided at two metropolitan public hospitals within Sydney Local Health District (SLHD) in New South Wales (NSW), Australia. The VFC (intervention group) is located at Royal Prince Alfred (RPA) Virtual Hospital (**rpa**virtual), while the in-person fracture clinic (control group) is situated at the RPA Hospital. **rpa**virtual is Australia's first virtual hospital established in February 2020 to enable patients to receive hospital-level care at home through virtual means (eg, video calls or remote monitoring), rather than visiting the traditional hospital for their healthcare needs.[10]

### Eligibility criteria
Patients referred to the VFC (eg, from local emergency departments, general practices or the in-person fracture clinic) will be identified and screened by a VFC physiotherapist and an orthopaedic doctor to determine if the patient is suitable for either model of care (virtual or in-person). The RECITAL study staff will contact eligible patients to invite them into this study. Figure 1 illustrates the trial design.

Patients will be invited to participate if they meet the following criteria:
► Have an acute (<6 weeks old) simple fracture that can be managed using a removable orthoses (eg, shoulder immobiliser, CAMboot or wrist splint).
► Aged ≥18 years.
► Have a condition that is deemed appropriate for virtual management by an orthopaedic doctor.
► Has access to a phone and will be within New South Wales at the time of consult.
► Is willing to participate and comply with the study requirements.
► Have a radiology scan or report to confirm the nature of the injury.

Patients will be excluded if they have:
► Complex or significantly displaced fracture, including pathological, open, unstable or spinal fractures requiring a cast or surgical management.
► Neurovascular concerns.
► A condition not managed by RPA Hospital Orthopaedics Department.
► Reported being unable to attend the in-person fracture clinic within the recommended follow-up time.
► Opted out.

People with any type of simple fracture that is deemed appropriate for virtual care will be eligible for the trial, to reflect usual practice. The most common types of fracture are expected to be base of fifth metatarsal, ankle Weber A, and Mason I radial head. Uncommon types of simple fracture could include greater tuberosity or clavicle. Patients who consent to participate and complete their baseline measures will be enrolled in this study. Informed consent (online supplemental file 1) and study data will be collected and managed using the Research Electronic Data Capture (REDCap) tool hosted at SLHD.[11 12] The randomisation schedule will be computer-generated using REDCap's randomisation model and will be stratified in random blocks of 4, 6, 8 and 10 to ensure equal numbers in both groups and concealed allocation. A biostatistician not involved in this study will set up the allocation schedule and upload it into REDCap. Only the biostatistician will be aware of the allocation to ensure concealment. The study coordinator will randomise the patients to the study groups. Participants randomised to the VFC who agree to participate in the qualitative

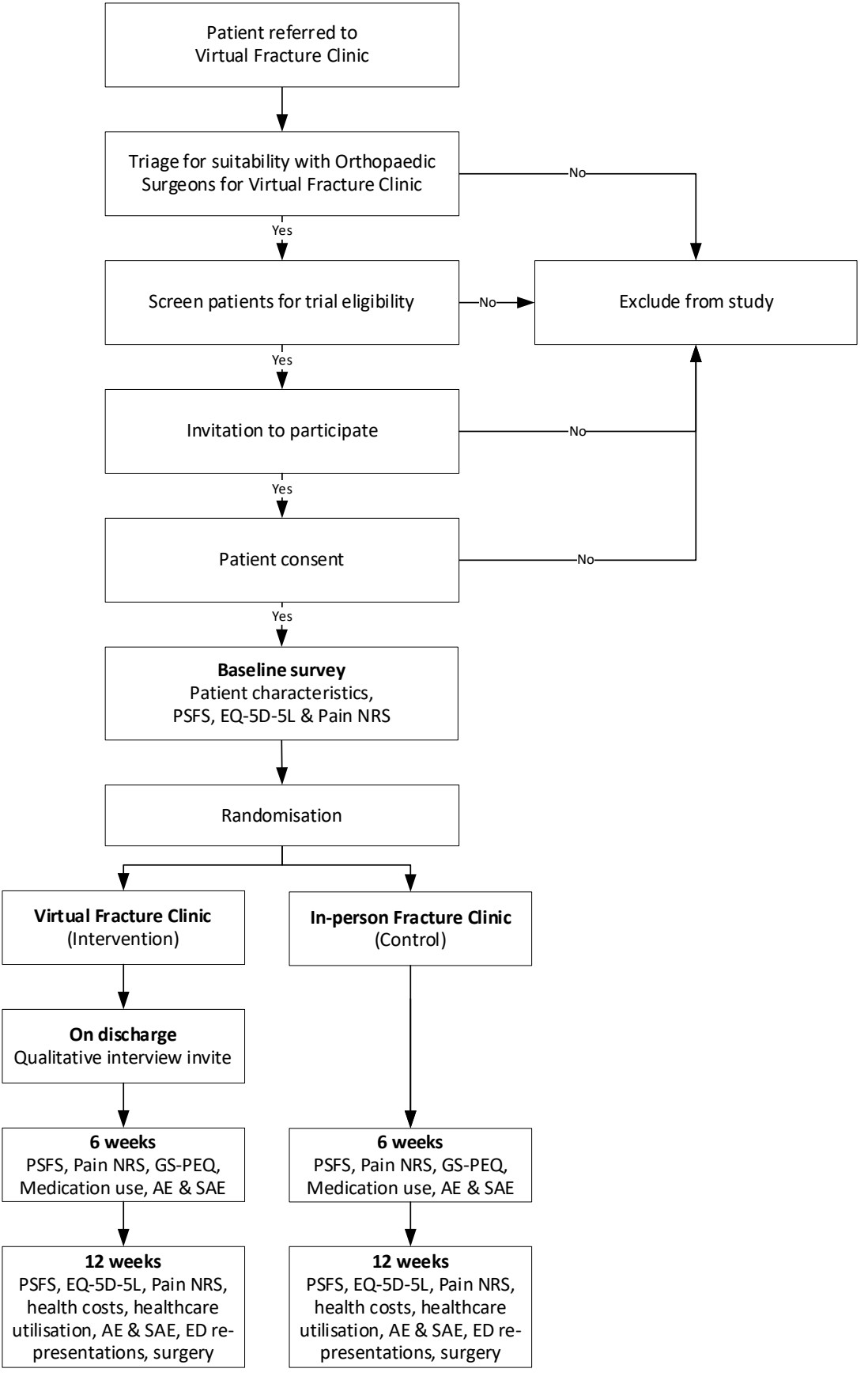

**Figure 1** Trial design. AE, adverse events; ED, emergency department; EQ-5D-5L, EuroQol 5 Dimensions 5 Levels; GS-PEQ, Generic Short Patient Experiences Questionnaire; NRS, Numerical Rating Scale; PSFS, Patient-Specific Functional Scale; SAE, serious adverse event.

substudy will be purposively selected for an interview according to their age, employment status, tertiary education level, type of injury and discharge status. Selected participants will be contacted once they are discharged from the clinic to ensure their complete experience with the VFC is captured.

### Interventions
Both study groups reflect current processes within existing clinics at the participating hospitals.

### VFC (intervention group)
Patients randomised to the VFC (intervention) group will be contacted via phone and email to organise an initial follow-up with a physiotherapist; usually within 5 days after their referral is received. Patients are sent an email with their appointment details and a fracture management fact sheet. The fact sheet explains their clinical condition, expected recovery, early rehabilitation exercises, activity limitations and information on care escalation. These fact sheets were adapted with permission from Royal Melbourne Hospital's VFC.

All patients are offered a video consultation with a physiotherapist unless they choose to have their review via phone. During the virtual consults, the physiotherapist conducts an assessment, discusses the X-ray findings and provides a management plan. The virtual consult sessions are usually approximately 30 min. An email summary of the consultation and follow-up appointment details are sent to the patient after the consultation. A Physitrack link may also be included in this email. Physitrack or PhysiApp is an internet-based programme that allows patients to view videos of their prescribed exercises. Patients are usually offered a follow-up virtual appointment at 2 weeks and 6 weeks post-fracture, or based on clinical need. Patients can contact the physiotherapist out of session if they have any concerns during their care period. Most patients are discharged from the VFC at 6 weeks post-fracture if there are no concerns. Patients will be supported with an interpreter, Aboriginal Cultural Support Team or with loaned devices and data as required. For example, although many older patients (aged 60+) currently use the virtual service, a digital patient navigator can assist patients and provide a smart phone with data so they can attend their virtual clinic appointments. We will monitor patient adherence by the number of consults attended; and the number of ad hoc patient contacts via phone or email with the clinic.

### In-person fracture clinic (control group)
Patients randomised to the in-person fracture clinic will be contacted via phone or email to provide a follow-up appointment. Appointments usually occur 7–10 days after the referral is received, based on the availability of the on-call orthopaedic doctor. Clinical management and subsequent follow-ups of the control group will be determined by the orthopaedic doctors at the in-person fracture clinic. Clinical management can include a physical assessment by a doctor, radiology scan, advice

and exercises. A physiotherapist may be involved in the patient's care. Patients in the control group may receive written instructions about their recovery and exercises as per current processes. The in-person consult sessions are usually approximately 20 min. Current practice suggests that patients may attend the in-person fracture clinic once or twice within 6 weeks post-fracture. We will monitor patient adherence by the number of consults attended.

Staff providing care to study participants will be trained on the trial protocol and be regularly supported by study investigators to ensure adherence to study protocol.

### Outcomes
The primary outcome of this study will be the participant's physical function assessed using the PSFS at 12 weeks. This self-reported tool has shown to be sensitive to change in patients with musculoskeletal problems, including simple fractures.[13–15] Participants list up to five functional tasks at baseline and score their level of ability — 0 (unable to perform activity) to 10 (able to perform activity at the same level as before the injury). Scores for each activity will be summed and calculated as an average of the total possible score for the participant (determined by the number of identified activities). We will compare the PSFS average scores between groups at 12 weeks as our primary measure. Table 1 summarises the outcome measures for this study.

Secondary measures for this study will include:
► Health-related quality of life measure assessed using the EuroQol 5 Dimensions 5 Levels (EQ-5D-5L) at baseline and 12 weeks.[16]
► Patient-reported experience measure assessed using the Generic Short Patient Experiences Questionnaire (GS-PEQ) at 6 weeks.[17]
► Pain assessed using the 0–10 Numerical Rating Scale at baseline, 6 and 12 weeks.[18]
► Cost borne by the healthcare service, measured at 12 weeks. We will collect data from the electronic medical records (eMR), SLHD Targeted Activity and Reporting System App Dashboard, and the hospital's performance, data and finance departments to obtain the healthcare appointment duration, healthcare provider's hourly rate, any health services utilisation and corresponding cost (including but not limited to outpatient, inpatient, ED, pharmacy, radiology, pathology and primary care), any infrastructure setup and maintenance cost, managerial and administrative overhead.
► Cost borne by patients, measured through the patient survey designed specifically for this study at 12 weeks.
► Healthcare utilisation assessed using a survey designed specifically for this study at 12 weeks. The survey will collect the number of other healthcare appointments for management of their injury. We will also ascertain if the patients used any other healthcare services through the patient's eMR.
► Medication use assessed using a survey designed specifically for this study to assess the name and dose

**Table 1** Outcome measures

| Milestones | Baseline | 6 weeks | 12 weeks |
|---|---|---|---|
| Patient characteristics | ✓ | | |
| Patient-Specific Functional Scale | ✓ | ✓ | ✓ |
| EuroQol 5D-5L | ✓ | | ✓ |
| Pain Numerical Rating Scale | ✓ | ✓ | ✓ |
| Generic Short Patient Experiences Questionnaire | | ✓ | |
| Health costs | | | ✓ |
| Healthcare utilisation | | | ✓ |
| Medication use | | ✓ | |
| Adverse events and serious adverse events | | ✓ | ✓ |
| Emergency department representations | | | ✓ |
| Patients requiring surgery | | | ✓ |
| EuroQol-5D-5L, EuroQol 5 Dimensions 5 Levels. | | | |

of prescription or over-the-counter medication for their injury at 6 weeks.

► Adverse events and serious adverse events assessed using a survey designed specifically for this study at 6 and 12 weeks. We will also collect data from safety reports within the eMR, and the NSW Health Incident Management System.

► ED representations measured by reviewing the eMRs at 12 weeks. This information may also be reported in the healthcare utilisation survey and the adverse/serious adverse event survey.

► Number of patients requiring surgery measured by reviewing the eMRs at 12 weeks. This information may also be reported in the healthcare utilisation survey and the adverse/serious adverse event survey.

### Sample size

A total sample of 312 participants will provide 90% power to detect a non-inferiority margin of 0.7 points on the 11-point PSFS with a 10% loss to follow-up, an SD of 2.0, α of 5% and a correlation score between baseline and final scores of 0.5 at 12 weeks.[19] A negative between-group difference of ≤0.7 points will indicate that the VFC clinic is non-inferior to the in-person fracture clinic.

We chose an SD of 2.0 as this is between the mean value of the SD for the PSFS at follow-up in published studies that range from 1.7,[20] 2.1[21] and 2.2.[13] The minimal important difference (MID) for the PSFS ranges from 1.3 (small change) to 2.7 (large change).[21 22] Guidelines suggest using a non-inferiority margin of 50% (or less preferably) of the treatment effect of standard care versus placebo.[23] Thus, we chose a between-group non-inferiority margin of 0.7 (50% of the MID of 1.3).

### Blinding

The participants, therapists and assessors will not be blinded. The surveys administered during this trial are self-assessments completed by patients directly in REDCap who will be blinded to the study hypothesis. If required, an independent blinded assessor may contact the patient to assist them with completing their surveys.

### Data collection methods

Patients will receive a unique link via email or phone message to complete all their surveys directly in REDCap. Patients will receive an email or phone message 2 days prior to each milestone, reminding them to complete their respective surveys. Two reminders followed by a phone call will be provided to patients who do not complete their surveys by the respective milestone. The treating clinicians may remind the participants to complete their surveys during their routine clinical reviews. The clinicians will not be able to complete, nor alter the results from the surveys. If requested, paper copies of the surveys may be sent to participants with their responses transcribed verbatim into REDCap by an independent blinded assessor not involved in this study.

### Data management

All study data will be collected, logged and stored within SLHD's REDCap server. REDCap functions such as adding a field note (a brief descriptor of the question or answer), autocalculations and using data validation functions will be used to ensure data quality. The 'required' field is also used to ensure participants complete the mandatory questions prior to submitting the survey. The questionnaires will be tested by clinicians and patients prior to implementation. The research team will have access through a personal login and password.

### Statistical methods

An intention-to-treat analysis will be implemented after the database is cleaned and locked. Separate analyses will be conducted on each outcome. Descriptive statistics will be used for patient demographics and clinical characteristics. Categorical variables will be described

with frequencies (%), and continuous variables will be described with means and SD. Data will be analysed using STATA V.14 statistical software (StataCorp) or R V.4.2.1 (R Foundation for Statistical Computing, Vienna, Austria. https://www.R-project.org/).

## Primary analysis

Non-inferiority trials assess whether an intervention outcome is not clinically worse than a control. The PSFS score at 12 weeks post-randomisation is the primary outcome in this study and we have prospectively defined a non-inferiority margin ($\Delta T$) of −0.7 points, which is the maximum difference we are prepared to tolerate and still consider virtual care not to be clinically inferior to in-person care. The null hypothesis is, therefore, that a difference of greater than $\Delta T$ exists in favour of in-person care (H0: $\Delta \leq -\Delta T$). This will be assessed by creating a 95% CI, which should be entirely above the non-inferiority margin for the intervention to be declared non-inferior. The PSFS score will be compared between treatment groups as the dependent variable in a generalised linear regression model for the primary analysis adjusting for baseline PSFS variables. The treatment difference will be based on the estimate of adjusted means and 95% CIs.

## Secondary analysis

Secondary clinical outcomes will be analysed using logistic regression for binary outcomes and linear regression for continuous outcomes. Results from the analyses will be presented as point estimates with 95% CIs. Baseline scores will be included in the model to increase statistical precision. If more than 5% of data are missing, then imputation techniques may be considered.

## Cost-effectiveness analysis

The economic evaluation will estimate the difference in the cost of resource inputs used by participants in the two arms of the trial, allowing comparisons to be made between the two models of care.

We will conduct a cost-effectiveness analysis to estimate the incremental-effectiveness ratios (ICERs) defined as: [cost of the virtual care−cost of in-person practice]/[effectiveness of the virtual care−effectiveness of in-person practice]. The effectiveness outcomes include PSFS, ED visit, rehospitalisation and quality-adjusted life-years (QALYs). Costs for resource inputs will largely be derived from available local and national sources and estimated in line with best practice. Primary research using established accounting methods may also be required to estimate unit costs. Costs will be standardised to current prices where possible. The EQ-5D-5L outcomes will be used to generate QALYs, and the responses will be compared with the national Australian value set for the EQ-5D-5L.[16] Multiple imputation methods will be used to impute missing data and avoid biases associated with complete-case analysis.

To estimate the uncertainty of ICERs, bootstrapping will be used to resample corresponding costs and effectiveness that will be observed in RECITAL, and the distribution of ICERs calculated from all resamples will be plotted on a cost-effectiveness plane. Subgroup analysis will be carried out to assess the equity impact of the interventions. One-way sensitivity analysis will be conducted around key cost variables. A cost-effectiveness acceptability curve will be plotted, which will provide information about the probability that an intervention is cost-effective, given the level of a decision maker's willingness to pay for each additional effectiveness outcome gained. The economic assessment method will adhere to the Consolidated Health Economic Evaluation Reporting Standards 2022.[24 25]

## Qualitative interview analysis

The thematic analysis will be based on Braun and Clarke's six-phase framework.[26] After the interview recordings are transcribed verbatim, the research team will independently annotate the transcripts to generate initial ideas and relevant phrases. A qualitative data analytic software (NVivo) will be used to code and organise the data into themes. The topic guide may be modified between interviews to enable new emerging themes from the interviews to be explored more in depth with subsequent patients. The data will be reported according to the Consolidated Criteria for Reporting Qualitative Research.

## Data monitoring

Given the relatively low-risk nature of the intervention, a data safety and monitoring board will not be used in this study. The study coordinator will provide feedback (at least once per year) to the investigator team, which consists of orthopaedic doctors, senior researchers, hospital executives and a consumer.

## Harms

Adverse and serious adverse events as defined by the National Health and Medical Research Council (NHMRC) will be monitored throughout this study.[27] Potential adverse events arising from this study include misdiagnoses or missed diagnoses; ED representations or surgical management of the fracture. All serious adverse events will be reported immediately to the investigator team and Human Research Ethics Committee.

## Auditing

There are no planned audits for this study.

## Consent or assent

The RECITAL study staff will contact all eligible patients to inform them about their follow-up options for their simple fractures using a standardised recruitment script. All patients who agree to have their follow-up care at one of the fracture clinics (virtual or in-person) will be invited to participate in this study. Study staff will send the patient a REDCap link via email or phone message for participants to view the study outline and requirements online, including the opportunity to download the participant information sheet. If the patient agrees to participate, they will complete an e-consent form within

REDCap. Participants who choose not to participate in the RECITAL study will be able to choose their follow-up at the virtual or in-person fracture clinic. Only patients who have completed the e-consent form will be enrolled into this study. Participants who complete all their surveys will be given A$50 to reimburse them for their time.

## Access to data

Only clinicians providing care to the participants and the study coordinator will have access to the identifiable data. All other investigators of this study will have access to the deidentified data. As per NHMRC requirements, the research data from this study will be retained for 15 years from the end of the trial.[28] Study protocol will be made available on reasonable request.

## Ancillary and post-trial care

Study participants are free to engage with other treatment providers such as their general practitioner or outpatient physiotherapist during and after this study for the management of their injury. These costs will not be borne by the study. This study will capture these visits to other healthcare providers for the management of this injury through the healthcare utilisation survey.

## Dissemination policy

The trial results will be submitted for publication in reputable international journals and will be presented at relevant professional conferences. The results will also be disseminated to the media. Authorship eligibility will align with the International Committee of Medical Journal Editors.

## Patient and public involvement

AD is a coinvestigator of this trial and has lived experiences at both fracture clinics investigated by this study. AD agreed that the research question was important and has reviewed and provided feedback on all the study documents. Facts sheets used by the VFC have been approved by the **rpa**virtual consumer group. This study will investigate the experiences of participants through the GS-PEQ and qualitative interviews. All participants can indicate on the consent form if they would like to receive the final study results.

## Conclusion

This trial has been designed to be embedded in usual clinical practice to evaluate two existing models of care at two urban public hospitals. Results from this trial will inform patients, clinicians, hospitals, policy-makers and health funders globally about the effectiveness of a VFC.

## ETHICS AND DISSEMINATION

Study has been approved by the Sydney Local Health District Ethics Review Committee (RPAH Zone) (X23-0200 and 2023/ETH01038; 30 June 2023). Any amendments to the trial protocol will require approval from the trial's steering committee and the ethics committee prior

to implementation. Recruitment commence in November 2023 and is expected to complete by September 2026.

**Author affiliations**
[1]Institute for Musculoskeletal Health, Sydney School of Public Health, The University of Sydney Faculty of Medicine and Health, Sydney, New South Wales, Australia
[2]RPA Virtual Hospital, Sydney Local Health District, Sydney, New South Wales, Australia
[3]Sydney Health Literacy Lab, Sydney School of Public Health, The University of Sydney Faculty of Medicine and Health, Sydney, New South Wales, Australia
[4]Department of Orthopaedic Surgery, Royal Prince Alfred Hospital, Sydney Local Health District, Sydney, New South Wales, Australia
[5]School of Public Health and Preventive Medicine, Monash University, Melbourne, Victoria, Australia
[6]Consumer Representative, Sydney, New South Wales, Australia
[7]University of New South Wales, The George Institute for Global Health, Newtown, New South Wales, Australia

**Contributors** ACT, CM and MJT conceived the idea for the trial. MJT drafted the manuscript and JZ, CM, ACT, INA, OH, MH, MJT, RL, KP, TC, IK, BW, MS, AD and JP contributed to the design of the study and critical review. All authors contributed to the design, intellectual content to the manuscript and approved the final version.

**Funding** This work is supported by the National Health and Medical Research Council (NHMRC) 2022 Medical Research Future Fund (MRFF) Clinician Researchers–Nurses Midwives and Allied Health (grant number 2022985) and Sydney Research 2022 Clinician Researcher Scholarship.

**Disclaimer** The funding sources did not contribute to the trial design and will not have a role in the trial conduct, data analysis, interpretation, writing or reporting.

**Competing interests** The following investigators (ACT, MS, OH, CM, TC, KP, JZ, INA, RL and AD) have no conflict on interests to declare. The SLHD clinicians (MJT, IK, BW, MH and JP) may deliver care to participants in either study groups as part of their usual clinical role provided at the public hospitals. MJT will be conducting this study in partial fulfillment of the requirements of a Doctor of Philosophy (Medicine and Health) degree under the supervision AT, CM, TC, KP and JZ.

**Patient and public involvement** Patients and/or the public were involved in the design, or conduct, or reporting, or dissemination plans of this research. Refer to the Methods section for further details.

**Patient consent for publication** Not applicable.

**Provenance and peer review** Not commissioned; externally peer reviewed.

**ORCID iDs**
Min Jiat Teng http://orcid.org/0000-0003-4948-0335
Joshua R Zadro http://orcid.org/0000-0001-8981-2125
Kristen Pickles http://orcid.org/0000-0002-1621-3217
Tessa Copp http://orcid.org/0000-0001-7801-5884
Ilana N Ackerman http://orcid.org/0000-0002-6028-1612
Christopher G Maher http://orcid.org/0000-0002-1628-7857
Adrian C Traeger http://orcid.org/0000-0002-1646-1907

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
