## [Reviewer comments · BMJ Open]

ARTICLE DETAILS

TITLE (PROVISIONAL)	RECITAL: A non-inferiority randomised control trial evaluating a virtual fracture clinic compared with in-person care for people with simple fractures
AUTHORS	Teng, Min Jiat; Zadro, Joshua; Pickles, Kristen; Copp, Tessa; Shaw, Miranda; Khoudair, Isabella; Horsley, Mark; Warnock, Benjamin; Hutchings, Owen; Petchell, Jeffrey; Ackerman, Ilana; Drayton, Alison; Liu, Rong; Maher, Christopher; Traeger, Adrian

VERSION 1 – REVIEW

REVIEWER	Jeyaraman, Madhan Sharda University
REVIEW RETURNED	23-Oct-2023

GENERAL COMMENTS	Well written manuscript VFC is a developing technology to treat and follow the fracture cases at a remote location Minor language polishing is mandatory
--

REVIEWER	Hutchison, Anne-Marie Swansea Bay Health Board
REVIEW RETURNED	01-Nov-2023

GENERAL COMMENTS	Overall a well designed study with very complex statistical design which involves aspects that are beyond our expertise. However we (there are 2 of us reviewing this together) feel there is one very early major flaw in the design in that specific fractures are not defined the authors have only suggest 4 examples of fracture types they would include and this suggests that others might be included. We feel that if the authors defined specific fractures only to include in the trial design (in other words the included fractures in the submission is a bit vague). Grouping too many fracture types together may give inappropriate power to some fractures that are less common and thus the results may be less valid.
--

REVIEWER	Buckley, R. Foothills Medical Centre
REVIEW RETURNED	24-Nov-2023

GENERAL COMMENTS	Good PRCT with a good question. Unclear as to how the older patient with limited computer adaptiveness or accessibility will be able to do a virtual visit. It should be stated clearly that this group of patients present a special challenge. Or, that group that has a problem with
--

	connectivity with that day, that place, that circumstance. Not every Zoom call works. Not everyone connects, no-shows happen. They should be acknowledged too with both sides of the PRCT. What about the time to get to a clinic, parking, hassle of an appointment? How is "convenience" measured? There must be a measure in the scientific world that will assist this study. These are simple fractures, this reviewer understands. But what about the person who has other questions about care like, wounds that need dressing, other areas of injury that are picked up late and that present to a clinic on the second trip and not the first visit that may need an xray or splinting? How will the virtual clinic deal with these? Is there any reimbursement for the patients in any way in this study? Be clear.
--	--

VERSION 1 – AUTHOR RESPONSE

Reviewer 1’s comments:

Minor language polishing is mandatory

Thank you. We have read through the protocol and made grammatical edits where required.

Reviewer 2’s comments:

Overall a well designed study with very complex statistical design which involves aspects that are beyond our expertise. However we (there are 2 of us reviewing this together) feel there is one very early major flaw in the design in that specific fractures are not defined the authors have only suggest 4 examples of fracture types they would include and this suggests that others might be included. We feel that if the authors defined specific fractures only to include in the trial design (in other words the included fractures in the submission is a bit vague). Grouping too many fracture types together may give inappropriate power to some fractures that are less common and thus the results may be less valid.

Thank you for your feedback. Our study aims to evaluate how well a virtual clinic could manage patients with acute simple fractures compared to an in-person fracture clinic. We have chosen to keep the types of included fractures general, instead of restricting it to certain conditions, as this is a pragmatic clinical trial embedded within current hospital services. The management of simple fractures is largely similar, involving immobilisation of the injured limb, providing advice on the recovery process, and starting early limb rehabilitation. We will list the details of participants’ fractures in a table for the final publication for readers to judge generalisability. We acknowledge that our study may not be sufficiently powered to determine subgroup effects, such as analysis for specific fracture types.

We have clarified this as follows:

Page 4 (Strength and limitations of this study):

- ‘Pragmatic clinical trial embedded within two existing fracture clinics at two urban hospitals
- Measures hospital-level outcomes as well as patient outcomes and experiences
- Blinding of therapist or participants is not possible, although participants are blinded to the study hypothesis
- Methods and results from this trial may inform the evaluation of other virtual musculoskeletal services
- Study may not be sufficiently powered to determine subgroup effects’

Page 7 (Eligibility criteria):

‘People with any type of simple fracture that is deemed appropriate for virtual care will be eligible for the trial, to reflect usual practice. The most common types of fracture are expected to be base of fifth metatarsal, ankle Weber A, and Mason I radial head. Uncommon types of simple fracture could include greater tuberosity or clavicle.’

Reviewer 3's comments:
Good PRCT with a good question.

1. Unclear as to how the older patient with limited computer adaptiveness or accessibility will be able to do a virtual visit. It should be stated clearly that this group of patients present a special challenge. Or, that group that has a problem with connectivity with that day, that place, that circumstance. Not every Zoom call works. Not everyone connects, no-shows happen. They should be acknowledged too with both sides of the PRCT.

The Virtual Fracture Clinic currently support patients (where required) with an interpreter, an Aboriginal Cultural Support worker, or a loaned device to reduce the inequity of accessing a virtual service. This was highlighted on Page 8 (Virtual Fracture Clinic (VFC) (Intervention Group)) and we have added some clarification:

'Patients will be supported with an interpreter, Aboriginal Cultural Support Team or with loaned devices and data as required. For example, although many older patients (aged 60+) currently use the virtual service, a Digital Patient Navigator can assist patients and provide a smart phone with data so they attend their virtual clinic appointments.'

2. Is there any reimbursement for the patients in any way in this study? Be clear.

We have added the following sentence to the protocol – Page 15 (Consent or assent):

'Participants who complete all their surveys will be given A\$50 to reimburse them for their time.'

Thank you.

VERSION 2 – REVIEW

REVIEWER	Hutchison, Anne-Marie Swansea Bay Health Board
REVIEW RETURNED	27-Dec-2023
GENERAL COMMENTS	We have only clicked N/A for the statistics question as we have recommended statistical review We are happy with the amendments to this study proposal from recommendations from our first review (second reviewer - Mr Andrew Bebbington FRCS -Consultant Trauma & Orthopaedic Surgeon)
REVIEWER	Buckley, R. Foothills Medical Centre
REVIEW RETURNED	21-Dec-2023
GENERAL COMMENTS	Good revision; no deficiencies noted.